# Early Pregnancy Markers in the Serum of Ewes Identified via Proteomic and Metabolomic Analyses

**DOI:** 10.3390/ijms241814054

**Published:** 2023-09-13

**Authors:** Yaying Zhai, Fan Xia, Luting Shi, Wenkui Ma, Xiaoyang Lv, Wei Sun, Pengyun Ji, Shuai Gao, Zoltan Machaty, Guoshi Liu, Lu Zhang

**Affiliations:** 1State Key Laboratory of Farm Animal Biotech Breeding, National Engineering Laboratory for Animal Breeding, College of Animal Science and Technology, China Agricultural University, Beijing 100193, China; zhaiyaying@outlook.com (Y.Z.); xiafanv587@163.com (F.X.); slt010112@163.com (L.S.); ma17610890927@163.com (W.M.); jipengyun@cau.edu.cn (P.J.); gaoshuai@cau.edu.cn (S.G.); gshliu@cau.edu.cn (G.L.); 2Frontiers Science Center for Molecular Design Breeding, China Agricultural University, Beijing 100193, China; 3Joint International Research Laboratory of Agriculture and Agri-Product Safety of Ministry of Education of China, Yangzhou University, Yangzhou 225009, China; dx120170085@yzu.edu.cn (X.L.); dkxmsunwei@163.com (W.S.); 4International Joint Research Laboratory in Universities of Jiangsu Province of China for Domestic Animal Germplasm Resources and Genetic Improvement, Yangzhou University, Yangzhou 225009, China; 5Department of Animal Sciences, Purdue University, West Lafayette, IN 47907, USA; zmachaty@purdue.edu

**Keywords:** ewes, proteomics, metabolomics, pregnancy markers, serum amyloid A, afamin

## Abstract

The diagnosis of ewes’ pregnancy status at an early stage is an efficient way to enhance the reproductive output of sheep and allow producers to optimize production and management. The techniques of proteomics and metabolomics have been widely used to detect regulatory factors in various physiological processes of animals. The aim of this study is to explore the differential metabolites and proteins in the serum of pregnant and non-pregnant ewes by proteomics and metabolomics. The serum of ewes at 21, 28 and 33 days after artificial insemination (AI) were collected. The pregnancy stratus of the ewes was finally determined through ultrasound examination and then the ewes were grouped as Pregnant (*n* = 21) or N on-pregnant (*n* = 9). First, the serum samples from pregnant or non-pregnant ewes at 21 days after AI were selected for metabolomic analysis. It was found that the level of nine metabolites were upregulated and 20 metabolites were downregulated in the pregnant animals (*p* < 0.05). None of these differential metabolomes are suitable as markers of pregnancy due to their small foldchange. Next, the proteomes of serum from pregnant or non-pregnant ewes were evaluated. At 21 days after AI, the presence of 321 proteins were detected, and we found that the level of three proteins were upregulated and 11 proteins were downregulated in the serum of pregnant ewes (*p* < 0.05). The levels of serum amyloid A (SAA), afamin (AFM), serpin family A member 6 (SERPINA6) and immunoglobulin-like domain-containing protein between pregnant and non-pregnant ewes at 21-, 28- and 33-days post-AI were also analyzed via enzyme-linked immunosorbent assay (ELISA). The levels of SAA and AFM were significantly higher in pregnant ewes than in non-pregnant ewes, and could be used as markers for early pregnancy detection. Overall, our results show that SAA and AFM are potential biomarkers to determine the early pregnancy status of ewes.

## 1. Introduction

Accurate and timely pregnancy diagnosis has a significant economic impact on sheep farming. Early pregnancy diagnosis is crucial to determine the pregnancy status of ewes, and can be used to evaluate the conception rate, the efficacy of artificial insemination (AI) and the presence of possible diseases. Ewes identified as non-pregnant can be rebred or inseminated, thereby shortening the lambing interval and providing an economic advantage to the producer. Furthermore, reproductive health is sensitive to toxic exposure, specifically to endocrine disruptor pollutants, and have long-term adverse effects.

At present, transabdominal ultrasonography is one of the most common early pregnancy detection methods. When applied to determine the pregnancy status of ewes on day 21, the sensitivity was reported to be as low as 44.4%, and its sensitivity reaches 100% only on day 35 of gestation and afterward [1]. Other available methods such as visual early pregnancy examination, transrectal ultrasonography, near-infrared spectroscopy, hormonal assays or radiography are also not ideal [2,3]. The pregnancy-specific protein B (PSPB) has been identified, and its protein size was determined to be between 47 and 53 kDa [4]. In addition, bovine pregnancy-associated glycoprotein (bPAG) from bovine fetal cotyledon has been purified to homogeneity by HPLC [5]. PAG is synthesized by mononuclear and binuclear cells of the trophectoderm of ruminants. A portion of PAG can be released into the maternal blood circulation and detected by PAG antibodies [6]. The concentration of PAG in cows increased after pregnancy, and there was a significant difference in PAG concentration between pregnant and non-pregnant cows on day 22 following insemination [7]. Test kits have been developed based on PAG and successfully used in the cattle industry. The molecular structure of bovine and sheep PAG are relatively similar. Although PAG-based early pregnancy determination methods for sheep have been tested [8,9,10], these test kits have not been widely used in sheep farms due to high costs or limited efficiency to identify early pregnancy status. Furthermore, the reproductive health of females is sensitive to nutrition, infectious diseases, toxic exposure—specifically to endocrine disruptor pollutants—and have long-term adverse effects [11]. An advanced early pregnancy diagnosis method would help famers to distinguish the ewes with subfertility. There is clearly a need to identify new biomarkers for early pregnancy detection in sheep.

Proteomic and metabolomic analyses are powerful tools to identify the dynamic proteins and endogenous metabolites in body fluids, which could provide a deeper understanding of the biological system at the molecular level [12,13,14]. Proteins and metabolites play a crucial role in regulating the molecular pathways associated with reproductive activities [15]. Proteomics and metabolomics are emerging high-throughput approaches which have enabled researchers to identify hundreds and thousands of molecular using a very low quantity of samples from males and females suffering from infertility [16,17]. Proteomic analysis was used to reveal potential early pregnancy or litter size-related biomarkers in pigs, goats, cattle and jennies [18,19,20,21,22,23,24]. More importantly, these techniques allowed researchers to explore the molecules involved in the medical conditions of pregnant females suffering from ectopic pregnancy, hypertension disorder or diabetes mellitus [25,26,27,28,29,30,31]. Research using metabolomics was also conducted to find biomarkers in serum to foretell litter size in sheep during 7 to 70 days of gestation, or metabolites such as FSH, P4, AMH and amino acids to predict the number of ovulated oocytes following ovarian super stimulation [32,33,34]. Thus, the latest approaches empowered us to identify potential biomarkers in the very early pregnancy stages of ewes.

Early and accurate pregnancy examination to shorten the lambing interval of ewes and improve the reproductive efficiency is of great economic significance for farmers. Therefore, we designed experiments to detect early pregnancy markers in ewes using the latest proteomic and metabolomic analyses. Serum samples from pregnant (21, 28 and 33 days post-artificial insemination) and non-pregnant ewes were collected and evaluated. We found that the levels of 29 metabolites and 14 proteins were significantly different between the experimental groups at 21 days post-AI. In addition, the levels of four proteins were examined via ELISA, and it was found that serum amyloid A (SAA) and afamin (AFM) levels were significantly elevated in pregnant compared to non-pregnant ewes on 21, 28 and 33 days post-AI. Our current study provides evidence that SAA and AFM could serve as a marker to advance the detection time and determine early pregnancy in ewes.

## 2. Results

### 2.1. Serum Collection and Pregnancy Diagnosis

At 21-, 28- and 33-days post-AI, blood samples of 30 ewes were collected and the sera were isolated for future examination. At 33 days post-AI, all 30 ewes were examined using a real-time B-mode ultrasound scanner. As presented in Figure 1A–D, 21 ewes were pregnant and nine ewes were not pregnant. The serum samples were identified as belonging either to the pregnant or non-pregnant group based on the ultrasound results and then were used for proteomic and metabolomic analyses.

### 2.2. Metabolomic Analysis of Serum

To evaluate the metabolomic changes in the sera of ewes on day 21 post-AI, six serum samples from pregnant (P) and non-pregnant (N) ewes, respectively, were examined using Liquid Chromatography Mass Spectrometry (LC-MS). The LC/MS raw data were processed using the Progenesis QI (Waters Corporation, Milford, CT, USA) software (version 2.2). The sample information, metabolite name and mass spectral response intensity were then exported and the metabolites were identified according to databases like HMDB (http://www.hmdb.ca/), accessed on 19 September 2022. There were 1355 shared metabolites in the sera of pregnant and non-pregnant ewes (Appendix A). For the positive ion, 803 metabolites were identified, and 772 of them were found in library and 446 metabolites in KEGG. Meanwhile, for negative ion, 549 metabolites were identified, 528 of which were found in library and 302 in KEGG. The correlation of the samples was then analyzed according to PLS-DA (Partial Least Squares Discriminant Analysis) using R Packages (Version 1.6.2). As presented in Figure 2A, the PLS-DA score chart showed that the degree of separation between groups P and N is significant. There were 11 unique metabolites in the pregnant group and 19 unique metabolites in the non-pregnant group (Figure 2B,C).

All the metabolites were checked with the KEGG and HMDB databases. As presented in Figure 2D, the most abundant compounds are phospholipids and lipids according to KEGG Compound Classification. Meanwhile, the lipid and amino acid metabolism also ranked among the top pathways based on KEGG Pathway analyses (Figure 2E). Similarly, when the names of the metabolites were checked in the HMDB database (Figure 2F), 492 compounds were identified as lipids and lipid-like molecules, which are compounds of the most abundance.

### 2.3. Differential Metabolites Detected between Pregnant and Non-Pregnant Ewes

The differential metabolites found between pregnant and non-pregnant ewes were further analyzed. As shown in Figure 3A, the levels of 20 metabolites in the sera of pregnant ewes were elevated and nine metabolites were reduced compared to that of non-pregnant ewes (listed in Table 1). The differential metabolites were then clustered according to biochemical pathways through metabolic enrichment and pathway analysis based on the KEGG database (Figure 3B,C). It was found that amino acid metabolism is ranked among the top pathways.

The expression profile of the 29 differential metabolites were then used to calculate variable importance in projection (VIP). As presented in Figure 3D, we found that indolophenanthridine,2-mercaptobenzothiazole and N-4acetylcytidine were the top three metabolites. Together with three additional ones, the level of six metabolites listed as their VIP values are presented in Figure 3E–J. It can be seen that the serum metabolism of sheep at different pregnancy statuses involves amino acid as well as lipid metabolism. Also, the fold changes of all differential metabolites are large enough for visual diagnosis kit development.

### 2.4. Proteomic Analysis of Serum

The proteins in the sera from three pregnant (P) and three non-pregnant (N) ewes 21 days post-AI were examined using data-independent acquisition (DIA) proteomics. Total protein annotation was then performed according to the uniport database and the functional information of proteins was comprehensively obtained. There were 321 proteins with functional annotation (Appendix A) and two unique proteins were detected in the serum of pregnant or non-pregnant ewes respectively (Figure 4A). Then, the KEGG pathway analysis was performed, which revealed that these proteins were enriched in 21 KEGG signaling pathways (Figure 4B). The subcellular localization of the proteins was also determined and 147 extracellular proteins and 84 cytoplasmic proteins were found (Figure 4C).

The differences in the level of proteins in the sera of pregnant and non-pregnant ewes were also analyzed. As shown in Figure 4D, three proteins were upregulated and 11 proteins were downregulated in the sera from pregnant ewes compared to those from the non-pregnant ones. The detailed information of the 14 differential proteins can be seen in Figure 4E and Table 2. As shown in Figure 4F, the results of the KEGG pathway enrichment analysis identified an increase in the activity of the signaling pathways related to maternal pregnancy, embryo implantation and embryonic development. Among the 14 differential proteins, the levels of serum amyloid A protein (SAA), afamin (AFM), serpin family A member 6 (SERPINA6), Ig-like domain-containing protein (ILDP), albumin, beta-A globin chain β, SMB domain-containing protein and vitronectin are presented in Figure 5A–H.

### 2.5. Six Levels of Serum Proteins in Pregnant and Non-Pregnant Ewes at Different Stages of Pregnancy as Determined using ELISA

To confirm the changes of proteins detected by proteomics, the levels of SAA, AFM, SPERIN6 and ILDP in serum samples from pregnant and non-pregnant ewes at 21-, 28- and 33-days post-AI were examined via an ELISA kit. The serum SAA in the pregnant ewes was significantly higher than that of the non-pregnant ewes at 21d (3.37 ± 0.22 vs. 2.70 ± 0.16 ng/mL), 28d (2.63 ± 0.43 vs. 1.99 ± 0.27 ng/mL) and 33d (3.55 ± 0.34 vs.1.96 ± 0.22 ng/mL) post-AI (Figure 6A,B). Similarly, the serum AFM in the pregnant ewes was significantly higher than that in the non-pregnant ewes at 21d (91.73 ± 4.91 vs. 79.72 ± 3.50), 28d (75.93 ± 5.09 vs. 66.22 ± 4.68) and 33d (94.26 ± 4.57 vs. 65.86 ± 3.61) post-AI (Figure 6A, B). The levels of SPERIN6 in the sera of pregnant ewes were significantly lower than that of non-pregnant ewes, only at 33 days post-AI. There was no difference in the serum ILDP levels between pregnant and non-pregnant ewes. Combined with the pathways enriched by the above differential proteins, SAA and AFM can be used as markers for the early pregnancy diagnosis of ewes.

## 3. Discussion

This study aimed to identify biomarkers for the early pregnancy diagnosis of sheep. The analysis of pregnancy-associated molecules has been the subject of many studies; using different species as the identification of such proteins offers advantages that include not only early pregnancy diagnosis, but also the prediction of litter size and the determination of the presence of reproductive diseases. Accurate and timely pregnancy diagnosis has a significant economic impact on sheep farming. In this set of experiments, the sera of ewes at 21-, 28- and 33-days post-AI were collected. The pregnancy status of ewes was confirmed via ultrasound 33 days post-AI. The serum samples from pregnant and non-pregnant ewes on 21 days post-AI were then used for metabolomic and proteomic examination. We identified 29 metabolites and 14 proteins differentially expressed between the two groups. However, none of the metabolites were suitable as markers for pregnancy due to the limited fold changes and no further investigation was conducted on these molecules. Among the 14 differentially expressed proteins, four proteins, including SAA, AFM, SPERIN6 and ILDP, were selected for ELISA detection. The results showed that the levels of two proteins (SAA and AFM) in the sera were significantly different between pregnant and non-pregnant ewes at 21, 28 and 33 days of gestation; we propose that these proteins can be used as protein markers to detect early pregnancy in ewes.

Knowing whether or not an ewe is pregnant soon after mating or artificial insemination is critical for sheep reproductive management. Numerous clinical and immunologic methods have been developed for sheep pregnancy diagnosis. Previously, methods based on ELISA had been used to assay hormonal, early pregnancy factor (EPF) and pregnancy-associated antigens in order to detect the early pregnancy of ewes [35,36,37,38]. However, the pregnancy diagnosis of ewes in practice still depends primarily on the identification of animals that do not return to estrus or the use of B-mode ultrasonography; these methods provide a pregnancy status from 35 days of pregnancy onwards. To shorten the interval between two successive inseminations and minimize the economic losses due to non-pregnant animals, new methods like the visual ELISA-PAG test, originally developed for cattle, have been tested on sheep; these tests proved highly accurate and efficient [8,35,36]. However, these immunological methods have not been widely used on sheep farms due to their high costs and complicated operation processes. Hence, new biomarkers and easy examination procedures for early pregnancy diagnosis are urgently required in the sheep industry. In this set of experiments, high-throughput metabolomic and proteomic methods were applied to detected unique molecules in the sera of pregnant and non-pregnant ewes. Metabolomics and proteomics are sensitive enough to detect thousands of biological molecules in one sample at the same time; they can be used to screen the relevant markers to detect early pregnancy. With this approach, we identified a total of 321 serum proteins in the two groups, among which 14 proteins were significantly different between pregnant and non-pregnant ewes. Four proteins showing obvious fold changes (SAA, AFM, ILDP and SPERIN6) were further analyzed via an ELISA kit. It was demonstrated that the level of SAA and AFM in the sera of pregnant ewes were significantly elevated since day 21 post-AI.

During early pregnancy, the female animal needs to adjust its endocrine system, immune system, metabolites and hormones to allow for the fetus to establish a connection with the maternal body [39]. The expression of many proteins’ changes in the female during the early stages of pregnancy. In our experiments, SAA and AFM proved to be suitable indicators of early pregnancy. When the body is injured and inflammation occurs, the SAA level increases and regulates the adhesion, migration, proliferation and aggregation of cells [40]. Interestingly, others found that elevated SAA levels may be related to abnormalities in the decidualization process, which may lead to maternal infertility or spontaneous abortion [41]. Therefore, further functional experiments are required to elucidate the role of SAA in the early pregnancy of ewes. The other protein, afamin (AFM), is a glycoprotein with vitamin E-binding properties and a putative function in fertility [42]. In previous clinical studies, afamin levels increased in the maternal serum during pregnancy and was proposed to be a potential predictor of preeclampsia [43,44]. Others also conducted a large number of studies on the relationship between serum AFM levels and inflammatory bowel diseases, tumor diseases, coronary heart disease and ovarian cancer [45,46]. AFM is a functional protein in humans, but its functions in ewes is still unclear.

In the current research, serpin family A member 6 (SERPINA6) was identified as another protein present at elevated concentrations in the sera of pregnant ewes. Currently, investigations regarding SERPINA6 focus on its function as a steroid-binding protein in the blood of mammals, reptiles, amphibians and birds [47,48]. In humans, the levels of cortisol in the plasma is associated with cardio-metabolic, inflammatory and neuro-cognitive traits or even diseases. It has been reported that a genetic variant of the SERPINA6 gene (rs7161521) is associated with diurnal and stress-induced hypothalamic–pituitary–adrenal (HPA) axis activity in children [48]. Meanwhile, other studies have found that a SERPINA6 gene variant encodes corticosteroid-binding globulin (CBG), a protein with corticosteroid-binding properties in blood [49,50]. However, the role of SERPINA6 in animals during pregnancy has not been studied. Thus, more research is needed to elucidate whether SERPINA6 is involved in the regulation of the pregnancy of ewes or its presence in serum is simply due to stress. Meanwhile, the level of an Ig-like domain-containing protein (accession number: A0A452FW61) was significantly reduced in pregnant ewes; the exact identity of this protein could not be determined due to technical limitations.

Taken together, the SAA and AFM proteins are differentially expressed between pregnant and non-pregnant ewes and can be used as biomarkers for early pregnancy detection in the sheep.

## 4. Materials and Methods

### 4.1. Animals and Experimental Design

The experiment was conducted using adult female Hu sheep (Ovis aries) at the experimental facilities of the Inner Mongolia Golden Grassland Ecological Technology Group Co., Ltd., Bayannur, China (latitude 40°13′ N, longitude 105°12′ E) in July 2022. All experimental protocols concerning the handling of animals were performed in accordance with the requirements of the Institutional Animal Care and Use Committee at the China Agricultural University.

### 4.2. Collection of Blood Samples from Ewes

Blood was collected from ewes at 21-, 28- and 33-days post-AI. After all the blood samples were placed in a coagulation-promoting tube for 2 h, they were centrifuged at 4000 rpm/s for 10 min. After centrifugation, the supernatant was collected and stored at a −80 °C freezer for later use.

### 4.3. Serum Metabolomic Analysis

#### 4.3.1. Serum Samples Selection and Quality Control

The sera with uniform color from 6 pregnant sheep (P1, P2, P3, P4, P5 and P6) and 6 non-pregnant sheep (N1, N2, N3, N4, N5 and N6) on 21 days post-AI were selected for metabolome analysis. A pooled quality control (QC) sample was prepared by mixing equal volumes of all samples. The QC samples were disposed and tested in the same manner as the analytic samples to represent the whole sample set, which was used to monitor the stability of the analysis.

#### 4.3.2. LC-MS/MS Analysis of the Serum Samples

The metabolomics analysis of the ewes’ serum samples was performed as described previously [51,52]. Briefly, 100 μL serum sample was added to a 1.5 mL centrifuge tube with 400 μL solution (acetonitrile/methanol = 1:1(*v*:*v*)) containing 0.02 mg/mL internal standard (L-2-chlorophenylalanine) to extract metabolites. The samples were mixed by vortexing for 30 s and then low-temperature sonicated for 30 min (5 °C, 40 KHz). The samples were placed at −20 °C for 30 min to precipitate the proteins. The samples were then centrifuged for 15 min (4 °C, 13,000× *g*). The supernatant was removed and dried under nitrogen and then re-solubilized with 100 μL solution and extracted via ultrasonication for 5 min (5 °C, 40 KHz), followed by centrifugation at 13,000× *g* and 4 °C for 10 min. The supernatant was transferred to sample vials for LC-MS/MS analysis. The LC-MS/MS analysis of the samples was conducted on a SCIEX UPLC-Triple TOF 5600 system equipped with an ACQUITY HSS T3 column (Waters, Milford, MA, USA) at Majorbio Bio-Pharm Technology Co., Ltd. (Shanghai, China). Data acquisition was performed with the Data Dependent Acquisition (DDA) mode.

#### 4.3.3. Data Preprocessing and Annotation

The raw data of LC/MS were preprocessed using Progenesis QI (Waters Corporation, Milford, CT, USA) software (version 2.2), and a three-dimensional data matrix in CSV format was exported. Internal standard peaks, as well as any known false positive peaks were removed from the data matrix and peak-pooled. At the same time, the metabolites were searched and identified based on database HMDB (http://www.hmdb.ca/), Metlin (https://metlin.scripps.edu/) and Majorbio database, accessed on 19 September 2022. In order to reduce the errors caused by sample preparation and instrument instability, variables with relative standard deviation (RSD) > 30% of QC samples were removed, and normalized as log10 logarithmization to obtain the data matrix for subsequent analysis.

#### 4.3.4. Differential Metabolites Analysis

The variance analysis on the matrix file were performed after data preprocessing. The R package (Version 1.6.2) performed orthogonal least partial squares discriminant analysis (OPLS-DA), and Student’s *t*-test and fold difference analysis were performed. The variable importance in projection (VIP) was obtained via the OPLS-DA model and the *p*-value was determined using the Student’s *t* test; the metabolites with VIP > 1, *p* < 0.05 were regarded as significantly different metabolites. All the metabolites were screened and differential metabolites among two groups were summarized and mapped into their biochemical pathways through metabolic enrichment and pathway analysis based on database search (KEGG, http://www.genome.jp/kegg/), accessed on 1 October 2022. The data were analyzed through the free online platform of Majorbio Cloud platform (cloud.majorbio.com), accessed on 1 October 2022.

### 4.4. Serum Proteomic Analysis

#### 4.4.1. Serum Samples Selection and Quality Control

The serum samples from 3 pregnant and 3 non-pregnant ewes at 21 days post-AI were selected. The serum proteins were extracted in fresh lysis buffer and 0.5% sodium deoxycholate with 1 × phosphatase inhibitor cocktail (PhosSTOP, Sigma-Aldrich, St. Louis, MO, USA). The protein concentration was measured via the BCA assay (Thermo Fisher Scientific, Waltham, MA, USA) and confirmed using Coomassie-stained short SDS gel as previously described [53,54]. Protein quantification was performed according to the kit protocol. After protein quantification, SDS-PAGE electrophoresis was performed.

#### 4.4.2. Proteolytic Hydrolysis and Peptide Quantification

The protein samples were taken and added with lysate, and then triethylammonium bicarbonate buffer (TEAB) was added at a final concentration of 100 mM. The iodoacetamide at a final concentration of 40 mM was added and reacted at room temperature for 40 min. Precooled acetone (acetone/sample *v*:*v* = 6:1) was added to each tube and kept at −20 °C for 4 h. The samples were then centrifuged at 10,000× *g* for 20 min and the pellets were collected. After trypsin digestion, peptide from the samples was drained via rotation vacuum concentration (Christ RVC 2-25, Christ, Erfurt, Germany). The peptide was extracted and re-dissolved with 0.1% trifluoroacetic acid (TFA). The supernatant was desalted with Sep-Pak C18 filter cartridge (Waters, Milford, MA, USA) and dried. Peptide quantification was performed according to the peptide quantification kit (Thermo Fisher Scientific, Waltham, MA, USA).

#### 4.4.3. DIA Mass Detection of a Single Sample

Based on peptide quantification results, the peptide samples were redissolved in spectrometry loading buffer (2% ACN with 0.1% formic acid), including appropriate iRT peptide, which was used to calibrate retention time, and were analyzed using an EASY-nLC system (Thermo Fisher Scientific, Waltham, MA, USA) with a timsTOF Pro2 mass spectrometer (Bruker, Bremen, Germany) at Majorbio Bio-Pharm Technology Co., Ltd. (Shanghai, China). Then, the data-independent acquisition (DIA) data were acquired using a timsTOF Pro2 mass spectrometer operated in DIA-PASEF mode. MS data were collected over an m/z range of 400 to 1200 and an ion mobility range of 0.57 to 1.47 Vs·cm^−2^. Both accumulation time and ramp time were set to 100 ms. During MS/MS data collection, each cycle contained one MS and ten PASEF MS/MS scans. Exclusion was active after 0.4 min. A total of 64 DIA-PASEF windows were used (25 Th isolation windows).

#### 4.4.4. Protein Identification

Spectronaut software (Version 14) was used to analyze the DIA-PASEF raw data based on the spectra library generated by DDA-PASEF data. Retention times were corrected by iRT and 6 peptides per protein and 3 ions per peptide were selected for quantitative analysis. The parameters are as following: Protein FDR ≤ 0.01, Peptide FDR ≤ 0.01, Peptide Confidence ≥ 99%, XIC width ≤ 75 ppm. The shared and modified peptides were excluded, and the peak areas were calculated and summed to generate the quantitative results.

#### 4.4.5. Statistical and Bioinformatics Analysis

The similarity and difference of proteins between different serum samples were compared via sample correlation and principal component analysis using R Packages software (Version 1.6.2) on Majorbio Cloud platform (https://cloud.majorbio.com), accessed on 1 October 2022. *p*-values were corrected for multiple testing using the Student’s *t*-test. The proteins with a *p*-value < 0.05 and |log2FoldChange| > 1 was considered as significantly differentially expressed proteins. GO enrichment and KEGG pathway enrichment analyses of these predicted target genes were performed using R Packages.

The differentially expressed proteins (DEPs) were identified as thresholds of fold change (>1.2 or <0.83) and *p*-value < 0.05. Functional annotation of all identified proteins was performed using GO (http://geneontology.org/) and KEGG pathway (http://www.genome.jp/kegg/), accessed on 1 October 2022. DEPs were further used for GO and KEGG enrichment analyses. Protein–protein interaction analysis was performed using the String v11.5.

### 4.5. Evaluation of Protein Levels in Sera of Ewes at Different Time Points

Blood samples of ewes at 21-, 28- and 33-days post-AI were collected and the serum from each sample was selected for analysis. The levels of SAA, AFM, SPERIN6 and ILDP in the samples were detected via ELISA kits purchased from Beijing Beijianxinchuangyuan Biotechnology Co., Ltd., Beijing, China. The analyses were performed according to the manufacture’s protocol. Briefly, 10 μL of the serum samples and 40 μL of diluent were added to the wells, and then the HRP-conjugate reagent was added to each well and incubated for 60 min at 37 °C. After that, 400 μL wash solution filled each well and this was repeated 5 times. Then, each well was filled with 50 μL chromogen solution A and 50 μL solution B, which were mixed gently and incubated for 15 min at 37 °C. Finally, 50 μL stop solution was added to each well. The Optical Density (O.D.) value of each sample was detected at 450 nm using a microtiter plate reader and the data were calculated based on the standard curve using the O.D. value for each well.

### 4.6. Statistical Analyses

The statistical analyses for proteomic and metabolomic analyses are described in each section. The levels of differentially expressed metabolites and proteins were evaluated using one-way ANOVA. Statistical significance at *p* < 0.05 is marked as * and at *p* < 0.01 is marked as **.

## 5. Conclusions

Early pregnancy diagnosis allows for optimizing the production and timely management of decisions, offering a greater reproductive output in sheep farming. In this manuscript, the researchers proposed to determine the unique molecules in the sera of pregnant ewes at 21 days post-AI. Altogether, 29 differentially expressed metabolites and 14 differentially expressed proteins were detected in the sera of pregnant ewes, compared with non-pregnant ones. The levels of four differentially expressed proteins were re-examined via ELISA, and it was found that the levels of SAA and AFM were significantly increased in the sera of ewes at day 21, day 28 and day 33 of gestation. Therefore, SAA and AFM can be used as markers of early pregnancy in ewes. Finally, precisely unraveling proteomic and metabolomic biomarkers and hallmarks related to the establishment of early pregnancy might be helpful for recognizing and improving the efficiency of post-implantation interplay between in utero-endometrial and placental anatomo-histological compartments and fetuses, following surgical or transvaginal/transcervical procedures used for the transfer of ex vivo-produced ovine and other mammalian embryos propagated by a variety of modern assisted reproductive technologies, such as somatic cell nuclear transfer and in vitro fertilization.

## Figures and Tables

**Figure 1 ijms-24-14054-f001:**
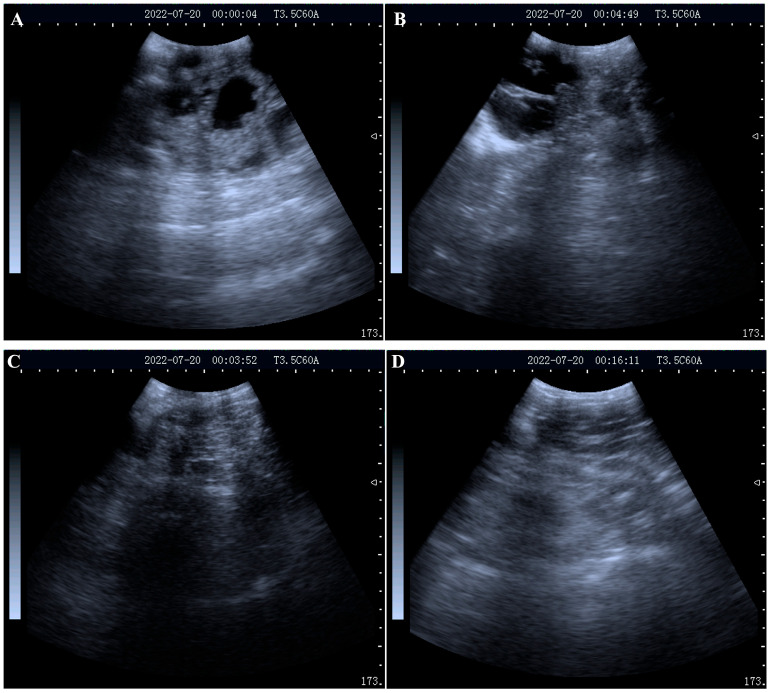
Representative images of the ultrasound examination: (**A**,**B**) ultrasound images of a pregnant ewe on day 33 post-AI; (**C**,**D**) ultrasound images of a non-pregnant ewe on day 33 post-AI.

**Figure 2 ijms-24-14054-f002:**
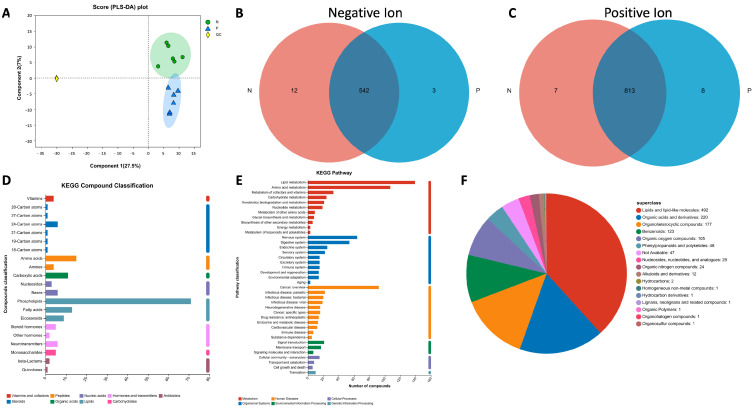
Analysis of the metabolites detected in the serum samples. (**A**) PLS-DA analysis score chart; (**B**,**C**) Venn diagrams for sample comparison. (**D**) Classification of metabolites based on KEGG Compounds analyses. (**E**) Classification of pathways based on KEGG. (**F**) Pie chart of compounds classification based on the HMDB database.

**Figure 3 ijms-24-14054-f003:**
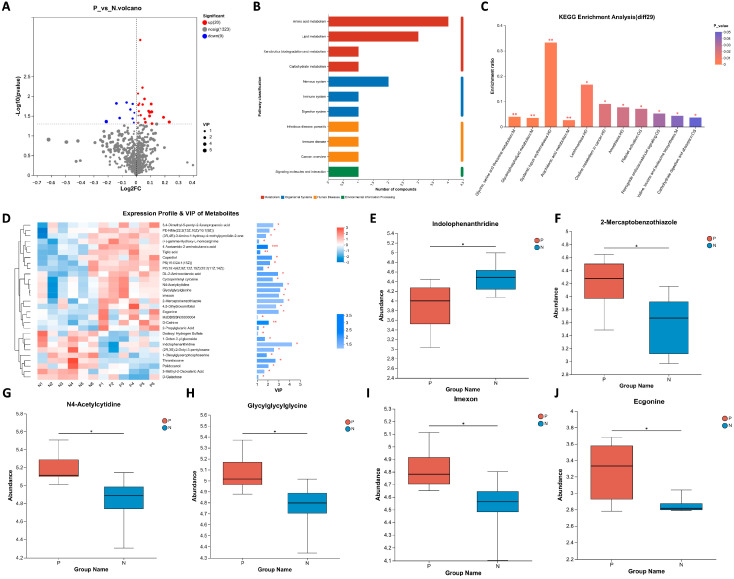
Analysis of the differential metabolites between pregnant and non-pregnant ewes on 21 days post-AI. (**A**) Volcano plot for differential metabolites. The red dots represent the upregulated metabolites and the blue dots represent the downregulated metabolites. (**B**) KEGG classification of the 29 differential metabolites. (**C**) KEGG pathway enrichment analyses of the 29 differential metabolites. (**D**) Expression profile of the 29 differential metabolites and their VIP values. (**E**–**J**) The differential metabolites ranked as the top 6 based on their VIP values. Statistical significance at *p* < 0.05 is marked as *, at *p* < 0.01 is marked as **, and at *p* < 0.005 is marked as ***.

**Figure 4 ijms-24-14054-f004:**
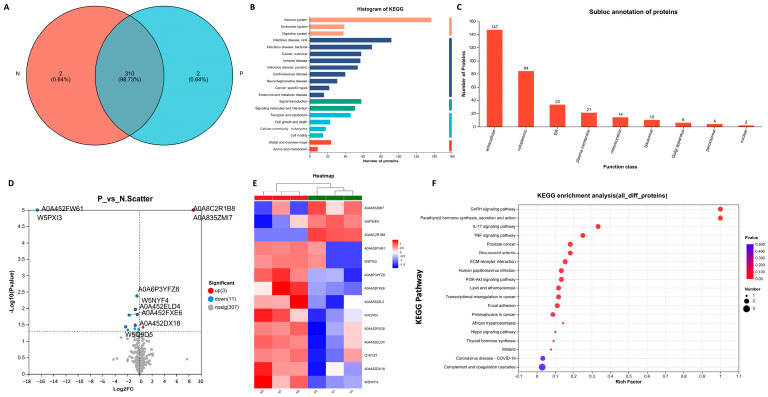
Proteomic analysis of sera from ewes. (**A**) Venn diagrams for the comparison of samples (P represents the pregnant ewes and N represents the non-pregnant ewes). (**B**) Total protein function annotation based on KEGG pathway enrichment analysis. (**C**) Subcellular localization analysis of the proteins. (**D**) Volcano plot of proteins detected in ewes. The red dots represent the upregulated proteins and the blue dots represent the downregulated proteins. (**E**) Heatmap of all the 14 differential proteins between the two groups. (**F**) KEGG pathway enrichment analysis for all the 14 differential proteins between the two groups.

**Figure 5 ijms-24-14054-f005:**
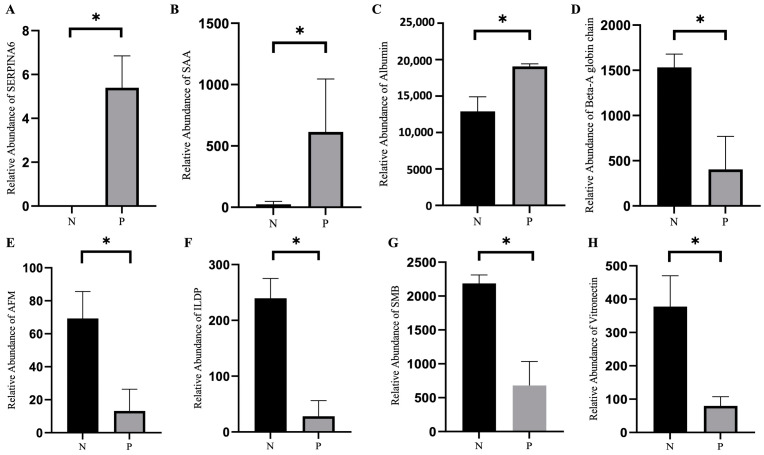
Levels of differentially expressed proteins in the sera of pregnant and non-pregnant ewes on 21 days post-AI. (**A**) Serpin family A member 6 (SERPINA6). (**B**) Serum amyloid A protein (SAA). (**C**) Albumin. (**D**) Beta-A globin chain β. (**E**) Afamin (AFM). (**F**) Ig-like domain-containing protein (ILDP). (**G**) SMB domain-containing protein. (**H**) Vitronectin. Statistical significance at *p* < 0.05 is marked as *.

**Figure 6 ijms-24-14054-f006:**
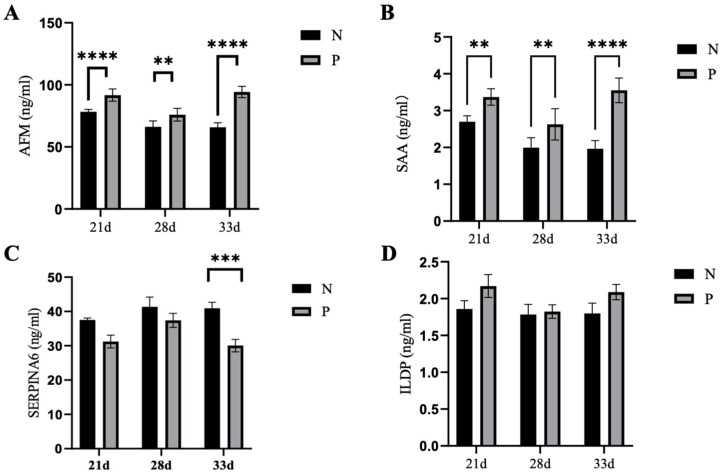
The level of proteins in sera from ewes at 21-, 28- and 33-days post-AI. (**A**–**D**): The levels of SAA, AFM, SPERIN6 and ILDP were compared at 3 different time points. Statistical significance at *p* < 0.01 is marked as **, at *p* < 0.005 is marked as ***, and at *p* < 0.001 is marked as ****.

**Table 1 ijms-24-14054-t001:** Differential metabolites between pregnant and non-pregnant ewes on day 21 post-AI.

Metabolite	Regulate	*p*-Value	FC (P/N)
Tiglic acid	up	0.008503	1.0083
4-Acetamido-2-aminobutanoic acid	up	0.0003752	1.0192
(3R,4R)-3-Amino-1-hydroxy-4-methylpyrrolidin-2-one	up	0.03014	1.0213
Glycylglycylglycine	up	0.02539	1.0668
N4-Acetylcytidine	up	0.02476	1.0778
3,4-Dimethyl-5-pentyl-2-furanpropanoic acid	up	0.04541	1.0456
DL-2-Aminooctanoic acid	up	0.01569	1.0753
Capsidiol	up	0.01163	1.0407
Polidocanol	down	0.02161	0.9714
4,5-Dihydrovomifoliol	up	0.04924	1.0791
PC(18:4(6Z,9Z,12Z,15Z)/20:2(11Z,14Z))	up	0.01659	1.01
1-Octen-3-yl glucoside	down	0.01561	0.9823
Indolophenanthridine	down	0.04359	0.8634
D-Cathine	up	0.006031	1.0305
Imexon	up	0.03142	1.0665
2-Propylglutaric acid	up	0.01513	1.015
1-Oleoylglycerophosphoserine	down	0.01433	0.9535
Thromboxane	down	0.01514	0.9073
(2R,3S)-2-Octyl-3-pentyloxane	down	0.0355	0.9217
PE-NMe(22:2(13Z,16Z)/16:1(9Z))	up	0.03717	1.0413
Dodecyl hydrogen sulfate	down	0.04624	0.9879
PS(15:0/24:1(15Z))	up	0.01613	1.0313
(+)-gamma-Hydroxy-L-homoarginine	up	0.0239	1.0124
2-Mercaptobenzothiazole	up	0.04465	1.1744
Ecgonine	up	0.03409	1.1418
MJDBISSN00000004	up	0.04705	1.0085
Cyclopentenyl cytosine	up	0.02362	1.0617
3-Methyl-2-oxovaleric acid	down	0.03653	0.9819
D-Galactose	down	0.02608	0.9901

**Table 2 ijms-24-14054-t002:** Differential proteins between pregnant and non-pregnant ewes on day 21 post-AI.

Accession Number	Description	Relative Abundance (P)	Relative Abundance (N)	FC (P/N)	*p*-Value (P/N)
A0A835ZZL2	SERPIN domain-containing protein	2825.4179	3124.8457	0.9042	0.04336
A0A452ELD4	SMB domain-containing protein	679.4033	2186.5892	0.3107	0.01607
A0A452DX18	Complement C3	864.3877	1408.2509	0.6138	0.03305
Q1KYZ7	Beta-A globin chain β	404.1686	1532.0683	0.2638	0.04558
W5Q9D5	Vitronectin	79.6122	377.7781	0.2107	0.03671
A0A6P3YFZ8	Uncharacterized protein	132.6988	176.5994	0.7514	0.004184
W5NYF4	Peptidoglycan recognition protein 2	56.3539	89.8142	0.6274	0.01077
A0A452FXE6	Insulin-like growth factor-binding protein 3	51.4896	66.5762	0.7734	0.01531
A0A452F0G8	Leukotriene A (4) hydrolase	9.4593	16.1675	0.5851	0.04163
A0A452FW61	Ig-like domain-containing protein	0	239.5522	1.00 × 10^−5^	0
W5PXI3	Afamin	0	69.3152	1.00 × 10^−5^	0
W5PWE9	Albumin	19,073.4655	12,902.1035	1.478	0.03812
A0A835ZMI7	Serum amyloid A protein	614.5201	0	424.2	0
A0A8C2R1B8	Serpin family A member 66	5.3973	0	424.2	0

## Data Availability

The raw data supporting the conclusions of this article will be made available by the authors, without undue reservation.

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
