# Peer review of "Early Pregnancy Markers in the Serum of Ewes Identified via Proteomic and Metabolomic Analyses"

_ijms, 2023, doi:10.3390/ijms241814054_

Round 1

Reviewer 1 Report

In this study, Zhai et al. aimed to investigate the selected metabolites and proteins present in the serum of pregnant and non-pregnant ewes using advanced techniques of proteomics and metabolomics. In this study, Zhai et al. aimed to investigate the selected metabolites and proteins present in the serum of pregnant and non-pregnant ewes using advanced techniques of proteomics and metabolomics. The authors demonstrate that two of the examined factors, serum amyloid A (SAA) and afamin (AFM), could become potential biomarkers for detecting the early pregnancy status of ewes. The manuscript submitted for review is exceptionally intriguing, and the obtained results may hold significance not only in terms of cognitive understanding but also potentially become a valuable diagnostic tool.

Addressing the below issues would substantially improve the manuscript:

M&M: I kindly request a more in-depth description of the analysis of selected proteins using ELISA kits. The current description is too concise, and there's also a lack of specification regarding the kits used.

The Conclusions section appears to be lacking the final paragraph focused on the implementation of the research results to the improvement of the efficacy for pregnancy establishment after transfer of embryos generated by in vitro production strategies. Therefore, it would be highly desirable if the following paragraph was added by the Authors:

"Finally, precisely unraveling proteomic and metabolomic biomarkers and hallmarks related to the establishment of early pregnancy might be helpful for recognizing and improving the efficiency of post-implantation interplay between in utero endometrial and placental anatomo-histological compartments and fetuses following surgical or transvaginal/transcervical procedures used for transfer of the ex vivo-produced ovine and other mammalian embryos propagated by a varietyof modern assisted reproductive technologies (ARTs) such as somatic cell cloning and in vitro fertilization (IVF) either by gamete co-incubation or by intracytoplasmic sperm injection (ICSI)."

Author Response

Dear reviewer,

We appreciate your comments and suggestions. The following are our responses.

In this study, Zhai et al. aimed to investigate the selected metabolites and proteins present in the serum of pregnant and non-pregnant ewes using advanced techniques of proteomics and metabolomics. In this study, Zhai et al. aimed to investigate the selected metabolites and proteins present in the serum of pregnant and non-pregnant ewes using advanced techniques of proteomics and metabolomics. The authors demonstrate that two of the examined factors, serum amyloid A (SAA) and afamin (AFM), could become potential biomarkers for detecting the early pregnancy status of ewes. The manuscript submitted for review is exceptionally intriguing, and the obtained results may hold significance not only in terms of cognitive understanding but also potentially become a valuable diagnostic tool.

Addressing the below issues would substantially improve the manuscript:

M&M: I kindly request a more in-depth description of the analysis of selected proteins using ELISA kits. The current description is too concise, and there's also a lack of specification regarding the kits used.

Response: Thanks for your suggestion. We added a detailed description of the ELISA procedure.

The Conclusions section appears to be lacking the final paragraph focused on the implementation of the research results to the improvement of the efficacy for pregnancy establishment after transfer of embryos generated by in vitro production strategies. Therefore, it would be highly desirable if the following paragraph was added by the Authors:

"Finally, precisely unraveling proteomic and metabolomic biomarkers and hallmarks related to the establishment of early pregnancy might be helpful for recognizing and improving the efficiency of post-implantation interplay between in utero endometrial and placent al anatomo-histological compartments and fetuses following surgical or transvaginal/transcervical procedures used for transfer of the ex vivo-produced ovine and other mammalian embryos propagated by a variety of modern assisted reproductive technologies (ARTs) such as somatic cell cloning and in vitro fertilization (IVF) either by gamete co-incubation or by intracytoplasmic sperm injection (ICSI)."

Response:  We appreciate your commons and this paragraph was added in the Conclusion of our manuscript.

Reviewer 2 Report

The aim of this manuscript is to explore the differential metabolites and proteins in the serum of pregnant and non-pregnant ewes by proteomics and metabolomics.

This manuscript shows rich content, providing a deep insight for some works: the study is within the journal’s scope, and I found it to be well-written, providing sufficient information. Even if the manuscript provides an organic overview, with a densely organized structure and based on well-synthetized evidence, there are some suggestions necessary to make the article complete and fully readable. For these reasons, the manuscript requires major changes.

Please find below an enumerated list of comments on my review of the manuscript:

ABSTRACT:

LINE 24: The authors should rewrite this sentence as following: “The aim of this study is to explore the differential metabolites and proteins in the serum of pregnant and non-pregnant ewes by proteomics and metabolomics”.

INTRODUCTION:

LINE 49: Furthermore, reproductive health is sensitive to toxic exposure, specifically to endocrine disruptor pollutants, and have long-term adverse effects (see, for reference: https://doi.org/10.3390/ijerph17072580). This manuscript may benefit from providing a brief and organic description of the effects of environmental pollutants on female and male reproductive competence.

LINE 70: Proteomics and metabolomics are emerging tools, to investigate the molecular mechanisms, which underlie the fertility (see, for reference: https://doi.org/10.1111/and.13711). The authors should highlight the pivotal role, played by proteomic and metabolomic analysis in the reproductive medicine, as suggested by recent evidence.

Furthermore, the authors should pay attention to the year of publication of this manuscript: not 2021, but 2023. Please, correct this information.

Finally, the authors should provide a list of the abbreviations, mentioned in the manuscript.

The main topic is interesting, and certainly of great clinical impact. As regards the originality and strengths of this manuscript, this is a significant contribute to the ongoing research on this topic, as it extends the research field on the differential metabolites and proteins in the serum of pregnant and non-pregnant ewes by proteomics and metabolomics. Overall, the contents are rich, and the authors also give their deep insight for some works.

As regards the section of methods, there is a specific and detailed explanation for the methods used in this study: this is particularly significant, since the manuscript relies on a multitude of methodological and statistical analysis, to derive its conclusions. The methodology applied is overall correct, the results are reliable and adequately discussed.

The conclusion of this manuscript is perfectly in line with the main purpose of the paper: the authors have designed and conducted the study properly. As regards the conclusions, they are well written and present an adequate balance between the description of previous findings and the results presented by the authors.

Finally, this manuscript also shows a basic structure, properly divided and looks like very informative on this topic. Furthermore, figures and tables are complete, organized in an organic manner and easy to read.

In conclusion, this manuscript is densely presented and well organized, based on well-synthetized evidence. The authors were lucid in their style of writing, making it easy to read and understand the message, portrayed in the manuscript. Besides, the methodology design was appropriately implemented within the study. However, many of the topics are very concisely covered. This manuscript provided a comprehensive analysis of current knowledge in this field. Moreover, this research has futuristic importance and could be potential for future research. However, major concerns of this manuscript are with the introductive section: for these reasons, I have major comments for this section, for improvement before acceptance for publication. The article is accurate and provides relevant information on the topic and I have some major points to make, that may help to improve the quality of the current manuscript and maximize its scientific impact. I would accept this manuscript if the comments are addressed properly.

Minor editing of English Language are required.

Author Response

Dear viewer,

We appreciate your comments and suggestions. The following are our responses.

The aim of this manuscript is to explore the differential metabolites and proteins in the serum of pregnant and non-pregnant ewes by proteomics and metabolomics.

This manuscript shows rich content, providing a deep insight for some works: the study is within the journal’s scope, and I found it to be well-written, providing sufficient information. Even if the manuscript provides an organic overview, with a densely organized structure and based on well-synthetized evidence, there are some suggestions necessary to make the article complete and fully readable. For these reasons, the manuscript requires major changes.

Please find below an enumerated list of comments on my review of the manuscript:

ABSTRACT: 

LINE 24: The authors should rewrite this sentence as following: “The aim of this study is to explore the differential metabolites and proteins in the serum of pregnant and non-pregnant ewes by proteomics and metabolomics”.

Response: Thank you for pointing this out.  As suggested by the reviewer, we have rewritten this sentence in the new version manuscript.

INTRODUCTION:

LINE 49: Furthermore, reproductive health is sensitive to toxic exposure, specifically to endocrine disruptor pollutants, and have long-term adverse effects (see, for reference: https://doi.org/10.3390/ijerph17072580). This manuscript may benefit from providing a brief and organic description of the effects of environmental pollutants on female and male reproductive competence.

Response: Thanks for your suggestion. We add sentences in Line 67-70 as “Furthermore, reproductive health of females is sensitive to nutrition, infectious diseases, toxic exposure, specifically to endocrine disruptor pollutants, and have long-term adverse effects [11]. An advanced early pregnancy diagnosis method would help famers to distinguish the ewes with subfertility.

LINE 70: Proteomics and metabolomics are emerging tools, to investigate the molecular mechanisms, which underlie the fertility (see, for reference: https://doi.org/10.1111/and.13711). The authors should highlight the pivotal role, played by proteomic and metabolomic analysis in the reproductive medicine, as suggested by recent evidence.

Response: Thanks for your suggestion. We add sentences in Line 76-80 as “Proteins and metabolites play a crucial role in regulating the molecular pathways associated with reproductive activities. The Proteomics and metabolomics are emerging high-throughput approaches which enabled the researchers to identify hundreds even thousand molecular using very low quantity of sample from male and female suffering infertility [14-15].”

Furthermore, the authors should pay attention to the year of publication of this manuscript: not 2021, but 2023. Please, correct this information.

Response: Thank you for pointing this out. We corrected this error.

Finally, the authors should provide a list of the abbreviations, mentioned in the manuscript.

Response: Thanks for your suggestion. We added a list of the abbreviations the revised manuscript.

The main topic is interesting, and certainly of great clinical impact. As regards the originality and strengths of this manuscript, this is a significant contribute to the ongoing research on this topic, as it extends the research field on the differential metabolites and proteins in the serum of pregnant and non-pregnant ewes by proteomics and metabolomics. Overall, the contents are rich, and the authors also give their deep insight for some works.

As regards the section of methods, there is a specific and detailed explanation for the methods used in this study: this is particularly significant, since the manuscript relies on a multitude of methodological and statistical analysis, to derive its conclusions. The methodology applied is overall correct, the results are reliable and adequately discussed.

The conclusion of this manuscript is perfectly in line with the main purpose of the paper: the authors have designed and conducted the study properly. As regards the conclusions, they are well written and present an adequate balance between the description of previous findings and the results presented by the authors.

Finally, this manuscript also shows a basic structure, properly divided and looks like very informative on this topic. Furthermore, figures and tables are complete, organized in an organic manner and easy to read.

In conclusion, this manuscript is densely presented and well organized, based on well-synthetized evidence. The authors were lucid in their style of writing, making it easy to read and understand the message, portrayed in the manuscript. Besides, the methodology design was appropriately implemented within the study. However, many of the topics are very concisely covered. This manuscript provided a comprehensive analysis of current knowledge in this field. Moreover, this research has futuristic importance and could be potential for future research. However, major concerns of this manuscript are with the introductive section: for these reasons, I have major comments for this section, for improvement before acceptance for publication. The article is accurate and provides relevant information on the topic and I have some major points to make, that may help to improve the quality of the current manuscript and maximize its scientific impact. I would accept this manuscript if the comments are addressed properly.

Response:  We appreciate your commons. As the reviewer suggested we added more description in the Introduction.

Round 2

Reviewer 2 Report

The authors have improved this manuscript. I accept for the publication of this paper.